# Forecasting the Regional Demand for Medical Workers in Kazakhstan: The Functional Principal Component Analysis Approach

**DOI:** 10.3390/ijerph22071052

**Published:** 2025-06-30

**Authors:** Berik Koichubekov, Bauyrzhan Omarkulov, Nazgul Omarbekova, Khamida Abdikadirova, Azamat Kharin, Alisher Amirbek

**Affiliations:** 1Department of Informatics and Biostatistics, Karaganda Medical University, Gogol St. 40, Karaganda 100008, Kazakhstan; omarbekova@qmu.kz (N.O.); harin@qmu.kz (A.K.); 2Department of Family Medicine, Karaganda Medical University, Gogol St. 40, Karaganda 100008, Kazakhstan; omarkulov@qmu.kz (B.O.); amirbek@qmu.kz (A.A.); 3Department of Physiology, Karaganda Medical University, Gogol St. 40, Karaganda 100008, Kazakhstan; abdikadirova@qmu.kz

**Keywords:** public health, healthcare workforce, FPCA, forecasting, time series

## Abstract

The distribution of the health workforce affects the availability of health service delivery to the public. In practice, the demographic and geographic maldistribution of the health workforce is a long-standing national crisis. In this study, we present an approach based on Functional Principal Component Analysis (FPCA) of data to identify patterns in the availability of health workers across different regions of Kazakhstan in order to forecast their needs up to 2033. FPCA was applied to the data to reduce dimensionality and capture common patterns across regions. To evaluate the forecasting performance of the model, we employed rolling origin cross-validation with an expanding window. The resulting scores were forecasted one year ahead using Autoregressive Integrated Moving Average (ARIMA) and Long Short-Term Memory (LSTM) methods. LSTM showed higher accuracy compared to ARIMA. The use of the FPCA method allowed us to identify national and regional trends in the dynamics of the number of doctors. We identified regions with different growth rates, highlighting where the most and least intensive growth is taking place. Based on the FPSA, we have predicted the need for doctors in each region in the period up to 2033. Our results show that the FPCA can serve as a significant tool for analyzing the situation relating to human resources in healthcare and be used for an approximate assessment of future needs for medical personnel.

## 1. Introduction

Healthcare workers play a central role in healthcare systems. This is evidenced by the COVID-19 pandemic worldwide, during which the most common reason for disruptions in the provision of essential health services was staff shortages [1].

The distribution of the health workforce affects the availability and accessibility of health service delivery to the public. The ideal distribution is presented as a fair distribution and availability of the health workforce across the population, regardless of geographic location. In practice, the demographic and geographic maldistribution of the health workforce is a long-standing national crisis.

In Kazakhstan, as elsewhere in the world, there are problems with staffing of healthcare organizations. One of the pressing issues is the shortage of medical personnel, especially in primary health care. According to government agencies, in 2023 the shortage of doctors in healthcare institutions amounted to 4864 full-time positions. Doctors bear a high workload, as evidenced by the underemployment rate of doctors, which stands at 1.4. An additional factor complicating the situation is the large proportion of people of retirement and pre-retirement age—9.5 thousand and 2.5 thousand people, respectively. Another serious problem is the uneven distribution of medical personnel, concentrated mainly in large cities, while economically underdeveloped regions and rural areas experience a shortage of qualified medical specialists of various profiles.

The Ministry of Health is actively introducing new human resource planning methods that consider the socio-economic, demographic, sanitary, and epidemiological characteristics of the regions. However, improving the system of planning and forecasting the provision of medical workers remains an urgent task [2,3].

This study attempts to mathematically model the annual needs of various regions of Kazakhstan for doctors (including general practitioners and specialists) up to 2033, based on Functional Principal Component Analysis (FPCA). The initial data for this analysis relate to the provision of doctors in 16 regions of Kazakhstan from 2000 to 2024.

FPCA is a statistical method that extends classical Principal Component Analysis (PCA) to functional data, such as curves or functions (e.g., time series or trajectories). FPCA reduces the dimensionality of functional data by identifying the main directions of variability (principal components) in the function space. FPCA provides a compact representation of complex functional data and highlights their key features [4,5]. Many real problems have been solved based on this method, including modeling the curvature of the human cornea [6], analyses of set of density curves where the argument variable is the logarithm of income [7], and fMRI scanning of regions of the central nervous system [8]. Other different applications of Principal Component Analysis to functional data have been developed, including modeling child and adolescent growth [9] and mortality and fertility rates [10], and identifying species from bloodstains [11]. The Principal Component Analysis method has been successfully applied to spatial anisotropy [12], interpreting lactate curves [13], analyzing kinematic data [14], and gene classification [15], exploring the major source of variations in glomerular filtration rate curves [16], and estimating mean and covariance for functional snippets [17]. Using this methodology, growth trajectories were determined to distinguish children with normal growth from those with poor growth [18]. In [19], the method is applied to data from participants in a family study of mood spectrum disorders to characterize differences in daytime mood patterns in individuals with major subtypes of mood disorders.

The FPCA approach was shown to provide a better estimate compared to other conventional methods to handle longitudinal data in biomedical applications [20,21,22,23,24,25].

Some works in the field of medicine and related sciences include the application of FPCA for the analysis of wastewater data to assess psychoactive substance use across 42 European cities. FPCA identified weekly temporal patterns of drug consumption, such as ecstasy (MDMA), using Fourier and B-spline basis functions. The method demonstrated high stability and robustness to missing data compared to traditional PCA and wavelet-based PCA [26]. A similar 2020 study examined the weekly consumption patterns of six different drugs in 17 Italian cities. FPCA extracted key functional principal components (FPCs), enabling a more accurate description of consumption dynamics [27].

In another study, FPCA was applied to analyze the logarithmic mortality rates of French males over 200 years. While not strictly medical, the B-spline-based methodology highlights FPCA’s potential for handling longitudinal medical data, such as vital signs or disease trajectories [28]. In related fields, FPCA has been used to analyze biomedical signals, such as electrocardiograms (ECG) and electroencephalograms (EEG). For instance, one study [29] employed FPCA to identify characteristic patterns in EEG data for epilepsy, improving diagnosis and patient monitoring.

FPCA has been used to study functional data in medical research, such as recovery trajectories after surgery. The method isolated key components of variability, simplifying the interpretation of complex time series [30]. Another study [31] used FPCA to analyze glucose changes in patients with diabetes to identify individual metabolic patterns and improve personalized treatment approaches.

Currently, there is no single ideal tool for planning human resources for medicine. Rather, there are several approaches, including modeling based on needs, demand, and supply, which apply various methods, such as regression models, simulation modeling, Markov chains, and others. In recent years, methods based on machine learning have been intensively developed. Each of these methods has its own advantages and disadvantages.

In our research, the FPCA method is used to analyze and forecast the needs of doctors in different regions of one healthcare system because it allows us to:-Identify subtle changes in regional patterns: the method can capture how the number of doctors in certain regions changes over time, including nonlinear trends that traditional models may miss.-Analyze regional shifts: the method allows you to track how the “peaks” in the number of doctors shift across regions (for example, from economically less developed regions to oil-rich regions or the capital), which may be due to an improvement in/worsening of the epidemiological situation or attractiveness. The method makes it possible to understand in which regions key changes are occurring.-Simulate long-term and short-term trends simultaneously: the functional approach includes high-order principal components, making it possible to capture complex time dynamics, such as slowdowns/accelerations or periodic fluctuations in the number of doctors associated with various reasons.-Analyze structural changes: the method can identify both general trends (for example, a general increase in the number of doctors in the Republic) and regional changes in individual regions.

The method supports coherent forecasting, where forecasts for different regions do not lead to unrealistic discrepancies or intersections. This helps the regulator create more plausible scenarios when planning human resources.

## 2. Materials and Methods

### 2.1. Data Sources

The dataset includes statistical data concerning Kazakhstan doctor counts by regions and by years in the period 2000–2024. The data for 2019 and 2020 were adjusted due to an unexplained decline in doctor counts in 2019, potentially reflecting reporting errors, followed by a correction in 2020. Interpolation between 2018 and 2021 values, along with manual adjustments for Region 8 (e.g., 1834 to 2090 in 2019), were applied to ensure data consistency.

These data were sourced from the Kazakhstan Bureau of National Statistics. According to the “Conditions for the use of official statistical information”, users without a concluded contract with the Bureau of National Statistics of the Agency on Strategic Planning and Reforms of the Republic of Kazakhstan may freely use official statistical information (including repeatedly) without charge, indefinitely, and without limitations with respect to the territory of use. This includes the rights to copy, publish, distribute with reference to the source, modify, and combine with other information, as well as use of the data for the creation of program products and applications (https://stat.gov.kz/en/description/, accessed on 1 October 2024).

Until 2017, Kazakhstan was divided into 16 first-level administrative-territorial units: 14 regions and 2 cities of republican significance. Subsequently, some of the larger entities were divided, and there are currently 20 of them. To ensure that the time series included a sufficient number of dimensions, we used the previous administrative division into 16 regions, as presented in Figure 1. The regions were ranked by decreasing population size in 2000 and assigned codes from 1 to 16.

As a result, the data set included observations from 2000 to 2024 in 16 regions of Kazakhstan. The real and corrected data set is presented in Appendix A.

### 2.2. Model Implementation

Functional Principal Component Analysis (FPCA) was applied to the data to reduce dimensionality and capture common patterns across regions.

The following assumptions are required to apply FPCA:The data can be represented as functional dependencies (regional and time profiles of doctor availability), which makes FPCA a suitable method.The data are pre-smoothed using orthogonal functions to represent them in functional form.It is assumed that a significant part of the variance in the data is explained by several principal components.The principal components can be interpreted in terms of national trends and regional features in the dynamics of the number of doctors.FPCA is similar to PCA and EFA, but is designed for functional data, which makes it the best choice in this case.The suitability of the method is also confirmed by the high accuracy of fitting and testing.

Let *y_t_*(*x*) be the observed doctor counts of region *x* in year *t*. It is assumed that it is realization of a smooth function *f_t_*(*x*) plus an observational error(1)$$y_{t}(x)=f_{t}(x)+\epsilon_{t}(x)$$

The time-dependent smooth function *f_t_*(*x*) is decomposed as(2)$$f_{t}(x)=\mu (x)+\sum_{j=1}^{J}\beta_{j}(x)\kappa_{j}(t)+e_{t}(x)$$
where:

*μ*(*x*) is the mean function of doctors counting for region *x* over time;

*β_j_*(*x*) is the set of orthonormal basis functions;

*κ_j_*,(*t*) is the set of time-varying coefficients (scores); and

*e_t_*(*x*) is the error term.

We used automatic extraction orthonormal basis functions directly from the data through FPCA. This ensures that the basis functions are adapted to the underlying structure of the mortality data (package *ftsa* in R4.4.2).

The modeling process consists of several steps:Set the *μ*(*x*) as the meaning of the *f_t_*(*x*) across the years.Find *β_j_*(*x*) and *κ_j_*(*t*) through a Principal Component Analysis and choose number *J* of them for the model.Choose a time series model for each of the *κ_j_*(*t*).Forecast as follows: assume the last year observed is *t* = *T*. The time series model for the *κ_j_*(*t*) provides us with *h*-step forecasts ${\hat{\kappa }}_{j}\left(T+h\right)$, which in turn give us the *h*-step forecasts(3)$${\hat{y}}_{T+h}\left(x\right)={\hat{f}}_{T+h}\left(x\right)=\hat{\mu }\left(x\right)+\sum_{j=1}^{J}{\hat{\kappa }}_{j}\left(T+h\right)\beta_{j}(x)$$

Two methods were used to forecast principal components scores:

ARIMA with automatic stationarity checking and best parameters selection [32].

LSTM model with a single LSTM layer (50 units), a dropout layer (0.2), and a dense output layer [33].

The choice of 50 units in the single LSTM layer of the model was determined based on a balance between model complexity and computational efficiency, tailored to the characteristics of the dataset and the forecasting task. The dataset, while informative, is relatively small in terms of temporal depth (25 data points per region), which imposes limitations on the capacity of the model to learn complex patterns without risking overfitting.

A moderate number of units, for instance 50, was selected to ensure the LSTM layer had sufficient capacity to capture the underlying temporal dependencies and trends while avoiding excessive parameterization that could lead to overfitting on this limited dataset. This choice aligns with empirical guidelines in time series forecasting with LSTM models, where the number of units is often set between 20 and 100 for small to medium-sized datasets, depending on the complexity of the patterns and the length of the input sequence (in this case, a look-back period of 3 years).

Preliminary experiments with alternative configurations (e.g., 30 and 100 units) were conducted during the model development. A model with 30 units showed insufficient capacity to capture the nuances of the trends, resulting in higher mean absolute percentage error (MAPE) values (approximately 4–5% on cross-validation). Conversely, a model with 100 units exhibited signs of overfitting, with MAPE improving on the training set but degrading on the validation set (MAPE > 4% on cross-validation for 2017–2024). The 50-unit configuration achieved a robust balance, yielding an average MAPE of 3.19% across regions during cross-validation, indicating a good generalization performance given the data constraints.

Additionally, the inclusion of a dropout layer (0.2) further regularized the model, mitigating the risk of overfitting associated with the 50 units. This architecture was deemed sufficient for the study’s objective of generating reliable forecasts for the number of doctors, as validated by the cross-validation results and the consistency of the predicted scores with historical trends.

The 50-unit single-layer LSTM was a practical and effective choice for this study given the dataset size and computational resources available.

The Adam optimizer was used with default settings (learning rate of 0.001), providing efficient and stable weight updates based on gradient descent. Preliminary tests with alternative optimizers (e.g., RMSprop) were conducted, but Adam demonstrated superior stability for the 25-year time series (2000–2024).

Mean squared error (MSE) was employed as the loss function, suitable for the regression task of forecasting the scores. MSE was selected due to its sensitivity to large deviations, which is critical for accurate predictions of doctor numbers where errors can have significant practical implications.

To ascertain the statistical significance of the FPCA–LSTM combination, a permutation test was implemented with 50 iterations. This involved randomly permuting the FPCA scores, retraining the LSTM model, and recomputing MAPE, allowing for the estimation of a *p*-value through comparison with the original MAPE distribution.

### 2.3. Model Validity

To evaluate the forecasting performance of the model, we employed rolling origin cross-validation with an expanding window [34]. The training set initially spanned the years 2000–2015 to forecast 2016, and with each subsequent fold, the training set was expanded by one year, up to 2000–2023 for forecasting 2024. For each fold, FPCA was performed on the training data. The resulting scores were forecasted one year ahead using ARIMA or LSTM. The forecasted scores were used to reconstruct the log-transformed number of doctors via the FPCA basis. The log-transformed forecasts were exponentiated to obtain the final forecasts.

This approach ensured that the temporal structure of the data was preserved while providing a robust assessment of the model’s predictive accuracy.

Model accuracy was evaluated using Absolute Percentage Error (APE):(4)$$APE=\left|\frac{A_{t}-F_{t}}{A_{t}}\right|\times 100\%$$
where *A_t_* is the actual data, *F_t_* is the forecast data at time *t*, and *n* is the number of forecast years.

Mean Absolute Percentage Error (MAPE)(5)$$MAPE=\frac{1}{n}\sum_{t=1}^{n}\left|\frac{A_{t}-F_{t}}{A_{t}}\right|\times 100\%$$

All the analysis in this study is performed using R.

### 2.4. Final Forecast for 2025–2033

The entire dataset (2000–2024) was used as the training period for the final forecast: FPCA was applied to the full log-transformed data, followed by forecasting of scores for 2025–2033. Monte Carlo Dropout was applied by performing 100 forward passes with Dropout enabled during inference. The key points for choosing the parameters were:-Bayesian Approximation: MC Dropout mimics Bayesian inference by sampling sub-networks, enabling uncertainty quantification.-Number of Passes: 100 passes balance computational cost and stable variance estimation, based on empirical guidelines.-Practical Benefits: improves robustness, quantifies uncertainty, and supports decision-making, especially for regions with variable prediction accuracy.-Implementation: dropout was re-enabled during inference, with predictions averaged over 100 samples.

The 95% confidence intervals were calculated for the reconstructed number of doctors.

## 3. Results

### 3.1. FPCA Decomposition of Region- and Time-Specific Doctor Counts

According to the FPCA results, the contribution of the first component is 94.7%, while for the second component it is 2% and for the third it is 1.3%.

The graphs in Figure 2 characterize national and regional trends in the doctor count by region (principal components β1, β2, β3) and by year (scores k1t, k2t, k3t).

The first principal component β1 within the FPCA framework represents the direction of maximum variation in the data, accounting for 94.7% of the total variance. In the context of our study, which analyzes the number of doctors (including general practitioners and specialists) across 16 regions from 2000 to 2024, β1 likely reflects the overarching long-term trend in doctor counts, dominating over regional or temporal fluctuations. This may be attributed to factors such as the overall growth of the medical workforce, driven by demographic changes, healthcare development, or national physician training programs.

In recent years, there has been a tendency towards an increase in the number of doctors in the Republic. The FPSA results also bear this out. For all regions except Region 8, the β1 values are positive, and the score k1t shows almost linear growth over the period 2000–2024 (Figure 2). The contribution of the first component (β1*k1t) of each region to the overall dynamics is presented in the graph in Figure 3. We have chosen to present this graph primarily to show the characteristics of regions 1, 8, 15, and 16 compared to other regions. Readers can view the full data for this graph in Appendix A (sheet PC1_contribution). Thus, the greatest positive contribution is made by Regions 15, 16, 1, etc. The number of doctors changes especially quickly in Region 15. In Regions 3, 4, 6, 9, and 10, the growth rates are lower, while in Region 8 there is a trend opposite to the general trend, i.e., in recent years there has been a decrease in the number of doctors.

The principal component β2 explains 2% of the variations, which is significantly less than that of the first component (94.7%) but still indicates some additional variation in the data that is not explained by β1. This means that β2 describes less pronounced, but still important patterns in the number of doctors by region and year.

The β2 values show how regions influence the overall trend (Figure 2). Positive and negative values indicate a contrast between regions. Positive β2 values for Regions 7, 13, and 14 indicate that these regions have similar trends. Negative values for Regions 3, 4, 5, and 15 indicate opposite trends compared to regions with a positive β2. Regions 8, 9, 10, and 11 have weights close to zero (e.g., −0.058, 0.056, 0.009, and −0.016), indicating their weak influence on β2. These regions probably do not show significant deviations from the overall trend. k2t shows how the second component changes over time (by year). Negative k2t values are most pronounced in the period 2000–2010 (e.g., −0.172 in 2008, −0.191 in 2016). That is, in these years, Regions 3, 5, 4, and 15 had a higher number of doctors compared to regions with a positive k2t (Regions 13, 14, and 12). Positive k2t values are observed in the years 2018–2025. In these years, regions with positive weights (Regions 13, 14, and 12) could have a relatively higher number of doctors compared to regions with negative weights. If you look at the contribution of the second component by region (Appendix A (sheet PC2_contribution)), you can see “nodal points”—years when the contribution in all regions was zero, i.e., the number of doctors in these years tended to the average value. These are the years 2000, 2007, 2011, 2014, and 2018.

The third principal component β3 and the score k3t are interpreted as factors representing residual temporal fluctuations or specific regional adjustments (Figure 2). Their contribution β3*k3t captures short-term or regionalized effects, such as epidemiological events or physician migration, which complement the national trend and regional peculiarities. For instance, the negative contribution of β3 in Region 8 suggests a decline in doctor numbers, while the positive contribution in Region 5 may indicate a unique growth not explained by general trends. Since the contribution of this component to the total variance is only 1.3%, the influence of these features is insignificant.

### 3.2. Model Accuracy Testing

To test the accuracy of the model’s forecasting, the entire time series of doctor counts in the regions was divided into two parts: training timeframe from 2000 to 2015 and testing timeframe from 2016 to 2024.

#### 3.2.1. Goodness of Fitting on the Training Timeframe

After the model parameters were calculated on the training timeframe, the doctor counts for 2000–2015 were recalculated, and the fitting data were compared with the real data. The mean absolute percentage error (MAPE) was used as a measure of fitting accuracy. The reconstructed data, incorporating the mean doctor counts across regions and years, achieved a MAPE ranging from 1.12% to 2.73%, with an average of approximately 1.76%, indicating a strong fit to the observed data (Table 1).

#### 3.2.2. Forecasting Accuracy on the Testing Timeframe

We tested the model for predictive capabilities in the medium term. For this purpose, we forecasted scores k1t, k2t, and k3t for 2016–2024 using One-Step-Ahead Forecasting based on the ARIMA and LSTM methods. The reconstructed time series values for these years were compared with real data and the MAPE metric was calculated for each region and each year (Figure 4).

For the ARIMA method, the MAPE across all regions and years is 3.69%, while for the LSTM method this figure is 3.19%. This suggests that LSTM is better at forecasting the number of doctors in general.

With LSTM forecasting, the range of average MAPEs across regions was from 0.81% (Region 3) to 6.99% (Region 5). The range by year was from 2.33% (2021) to 4.18% (2020). With ARIMA forecasting, the range of average MAPEs across regions was from 1.50% (Region 7) to 9.94% (Region 15). The range by year was from 2.79% (2024) to 4.41% (2018). LSTM is more stable, since its errors vary less both across regions and across years. LSTM has smaller peak errors (12% vs. 17.7%), making it more robust for regions with anomalous dynamics.

The advantages of LSTM were evident when analyzing data from Region 15. These regional data, covering the years 2000–2024, show a strong upward trend in the number of doctors, from 2198 doctors in 2000 to 10,540 in 2024, with an average annual growth rate of approximately 347.58 doctors per year. However, this trend is interrupted by rare but significant anomalies. In 2010–2011, the number of doctors declined from 5488 in 2000 to 5417 in 2011. This is a rare event for the region, which otherwise shows monotonic growth. After the 2011 decline, the region saw a sharp increase of 812 doctors in 2012 and another large increase of 665 doctors in 2014. These spikes are outliers compared to the average annual rate. With LSTM, the MAPE for Region 15 is 5.41%, which is significantly better than with ARIMA (9.94%). LSTM is generally preferred as it shows lower average error and more stable results.

The permutation test yielded a mean MAPE of 26.01% for permuted data, with a 95% confidence interval of [19.79%, 30.58%], reflecting a substantial increase in error when temporal patterns were disrupted. The resulting *p*-value of 0.000 confirmed the statistical significance of the model, suggesting that the FPCA–LSTM approach effectively captures meaningful patterns beyond random variation. These findings underscore the robustness of the hybrid methodology for time-series forecasting in healthcare resource allocation.

### 3.3. Forecasting Doctors Count to 2033

Based on the test results, we used the model to forecast the number of doctors for the years 2025–2033. In this case, the training timeframe included data for the years 2000–2024. Then, the scores were forecast for the period 2025–2033 by LSMT (Figure 5).

The first principal component score indicates that the general trend of an increasing number of doctors in the regions will continue in the forecast period. This trend will continue to be most strongly associated with growth in Regions 15, 16, and 1 (Figure 6, contribution β1). The contribution of Region 8 will remain negative.

The second principal component score, characterizing regional features, shows that in different regions the expected changes will have different dynamics. Thus, in Regions 1, 2, 6, 7, 9, 11, 12, 13, 14, and 16, in 2025–2033 the growth rate of the number of doctors will decrease, while in other regions the growth rate is expected to increase (Figure 6, contribution β2). The greatest acceleration will be observed in Region 15. In Region 8, the model predicts stagnation.

The predicted values of doctor counts for each region and each year are presented in Figure 7 and Appendix A (sheet Doctors_Forecast_2033).

## 4. Discussion

In the field of public health, FPCA studies mainly focus on the analysis of mortality, fertility, migration processes, and population size. However, the health workforce also exhibits spatial and temporal patterns. About 80% of health services in Kazakhstan are provided in the public sector. Almost half of medical students are funded by state grants. Therefore, government agencies have powerful tools to regulate the balance between the supply and demand of health resources both in the country as a whole and in individual regions. However, the problem of staff shortages and imbalances remains relevant. The growing population, the growth of chronic diseases, uneven socio-economic development of different regions, urbanization, and insufficient incentives to attract health personnel to rural areas all increase the requirements for planning and forecasting processes to meet health challenges.

Statistical and machine models (e.g., ARIMA, linear regression, neural networks) are built on historical data: they identify trends, seasonality, cycles, and correlations. The use of the FPCA method in our study allowed us to analyze medical personnel in 16 regions simultaneously while at the same time identifying national and regional trends in the dynamics of the number of doctors. We identified regions with different growth rates, showed where the most intensive growth is taking place (1, 15, 16), and identified a region in which the dynamics do not coincide with the general trend, namely, Region 8.

If the purpose of the method is forecasting, then after the stage of calculating the principal components and their coefficients, the stage of extrapolation of these scores to the forecast period begins. A wider range of univariate time series models may be used for this purpose. In this study, we used the ARIMA and LSTM methods.

While most regions were successfully modeled using ARIMA, Region 15 consistently exhibited high forecast errors. Region 15 exhibits dynamics that are out of sync with other regions. As noted, this region, the new capital of the Republic, is characterized by rapid population growth and, accordingly, increasing needs for medical personnel. The observed combination of a strong upward trend with rare anomalies (e.g., the 2011 decline) and a slowdown after 2015 made the dynamics of Region 15 less consistent with the general patterns captured by FPCA. This inconsistency led to a poor representation of Region 15 in the FPCA space, contributing to high forecast errors.

FPCA, which aims to identify general patterns across regions using a small number of principal components (in this case, three), tends to smooth out such rare anomalies. The model likely interpreted these events as noise rather than as region-specific phenomena, leading to systematic forecasting errors for Region 15. For example, during cross-validation (2016–2023), the model may not have accounted for the impact of these anomalies on the overall trend, leading to a high MAPE.

At the model testing stage, LSTM showed higher accuracy compared to ARIMA. LSTM is a type of recurrent neural network (RNN) specifically designed to work with time series. It can capture complex nonlinear dependencies in data due to its architecture with memory cells and gates (forget, input, output gates). LSTM was able to better model nonlinear dependencies inherent in some regions. In general, LSTM was more stable and robust.

Based on these results, we estimated the need for doctors by regions of Kazakhstan until 2033. It is expected that, in the medium term, the trend towards an increase in the number of doctors will continue in the Republic as a whole and in most regions. The leaders in demand will still be the new capital, the southern densely populated region and the western oil and gas region. The demand for doctors is predicted to remain unchanged in Region 8 and to decrease in Region 6.

## 5. Conclusions

As is the case around the world, the demand for medical specialists in Kazakhstan outpaces their supply. Young workers are flocking to large cities and industrial centers, while the imbalance between different regions is becoming more acute. Our findings may help regulators address staffing issues. Despite the importance of the results obtained, our study is not without certain limitations:-Our calculations were based on real data of doctor counts in the regions of Kazakhstan. However, in all the analyzed years, there was a shortage of personnel in the country. Unfortunately, we do not have access to information on the number of vacancies by region in these years. Taking this information into account would allow us to obtain more accurate forecasts.-The need for medical personnel is influenced by various socio-economic, demographic, and epidemiological factors. Taking these factors into account in FPCA is possible but difficult, due to the uncertainty of their impact in the future. In this case, the use of scenario methods is justified.-In our study, we predicted the total number of doctors in the regions, including general practitioners and specialists. More valuable information for regulatory authorities is the forecast of doctors by specialty. This allows for the efficient allocation of resources for training specialists.

The Ministry of Health of the Republic pays considerable attention to the problems of planning and forecasting medical personnel. Currently, methods are being introduced that take into account the socio-economic, geographical, and demographic characteristics of different regions of Kazakhstan. Our results show that the FPCA can serve as a significant tool for analyzing the situation with human resources in healthcare and can be used for an approximate assessment of future needs for medical personnel. Further development of the method with the inclusion of various external factors in the model will create a real basis for its use in the practical activities of regulatory authorities. Separate planning of medical personnel in rural and urban areas is of interest, since in sparsely populated rural areas the number of doctors cannot be determined only by the population size. It is also important to predict a possible shortage of doctors by region in order to begin training young medical specialists today. Future models should take into account the process of transferring a number of functions from a doctor to a nurse, which is gaining momentum in Kazakhstan.

## Figures and Tables

**Figure 1 ijerph-22-01052-f001:**
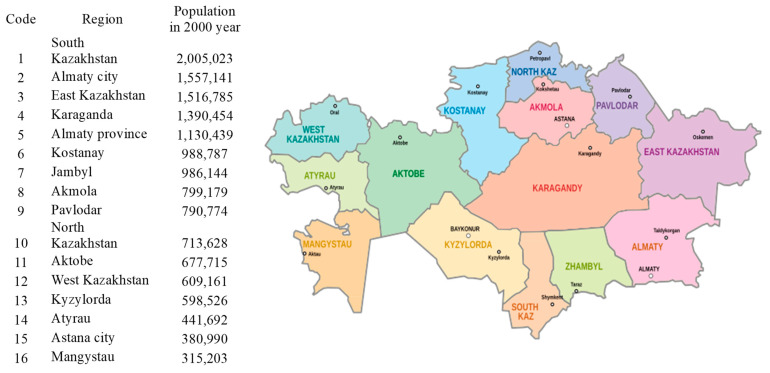
Kazakhstan regions encoding.

**Figure 2 ijerph-22-01052-f002:**
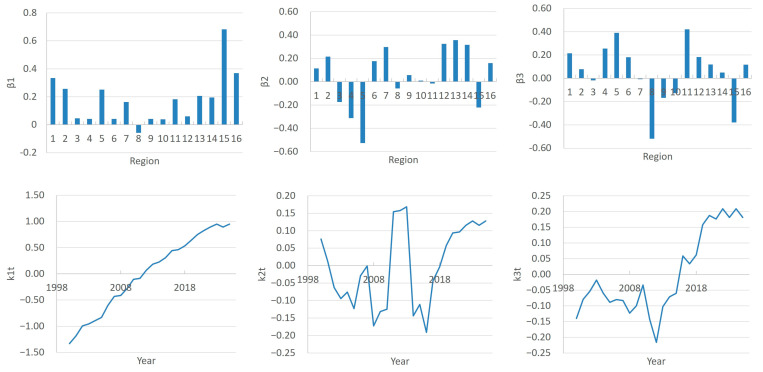
FPCA parameters.

**Figure 3 ijerph-22-01052-f003:**
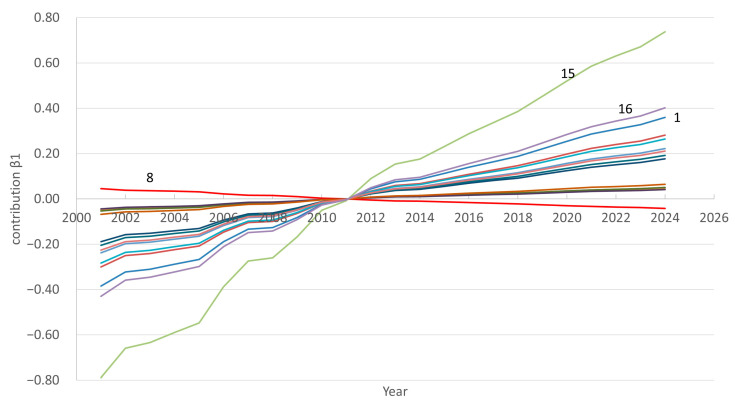
Contribution of the first principal component by region.

**Figure 4 ijerph-22-01052-f004:**
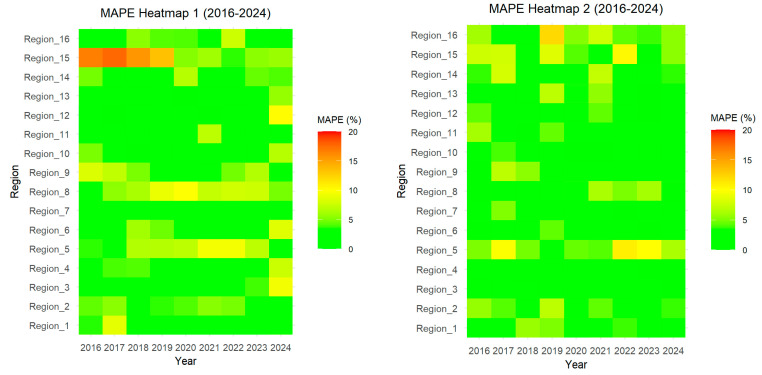
Model prediction accuracy of the ARIMA (Heatmap 1) and LSTM (Heatmap 2) techniques.

**Figure 5 ijerph-22-01052-f005:**
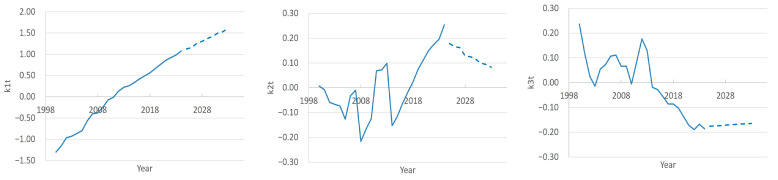
Principal component score forecasting by LSTM up to 2033.

**Figure 6 ijerph-22-01052-f006:**
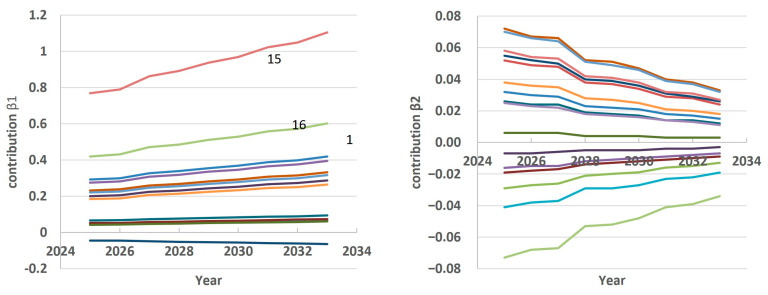
Contribution of the first and second principal components to the total variability.

**Figure 7 ijerph-22-01052-f007:**
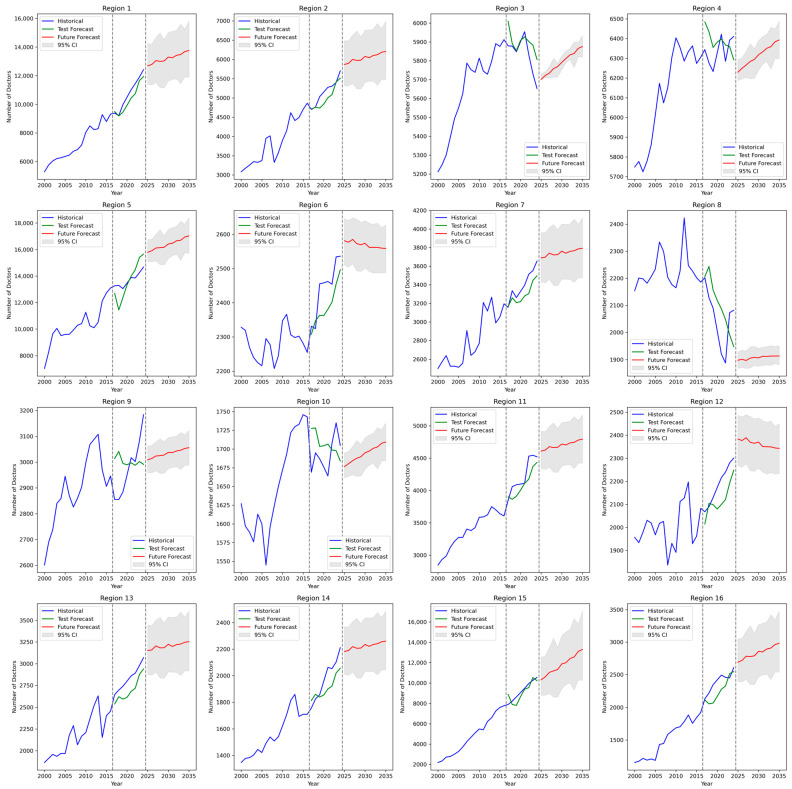
Prediction of regional demand for doctors.

**Table 1 ijerph-22-01052-t001:** Fitting results by regions and years.

Year	MAPE	Region	MAPE
2000	1.98	1	2.49
2001	1.31	2	2.73
2002	1.56	3	1.86
2003	1.15	4	1.12
2004	1.37	5	1.63
2005	2.02	6	1.34
2006	2.58	7	2.14
2007	2.31	8	1.8
2008	1.21	9	2.14
2009	1.12	10	1.43
2010	1.6	11	1.54
2011	2.67	12	1.58
2012	1.38	13	1.95
2013	1.91	14	1.23
2014	2.09	15	1.22
2015	1.87	16	1.94
Mean	1.76		1.76

## Data Availability

The data presented in this study are available on the websites of Kazakhstan national statistics bureau https://stat.gov.kz (accessed on 1 October 2024).

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
