# Peer review of "Forecasting the Regional Demand for Medical Workers in Kazakhstan: The Functional Principal Component Analysis Approach"

_ijerph, 2025, doi:10.3390/ijerph22071052_

Round 1
Reviewer 1 Report
Comments and Suggestions for Authors
- Overall this paper has a good purpose and findings. The abstract is well explained. However please include a brief conclusion of the study in the Abstract section. Please remove the word “It is expected” (line 25) in the abstract section. Also, please remove the word “humane resource” (line 28) and “analyses” (line 29) in the Keywords section.
- The introduction part could be improved with adding the recent studies that related to the “Functional Principal Component Analysis” (line 53).
- The data sources and model implementation are well explained. However, please write the mathematical notation carefully (line 113, 114,116, etc). It should be written using Times New Roman and Italic fonts. In addition, please explain why LSTM model used 50 units in the study (line 134). Also, please explain why Monte Carlo Dropout was applied by performing 100 forward passes with Dropout enabled during inference (line 157).
- My suggestion is to combine “Results” and “Discussion” in one section. Please elaborate the results for k2t, B3 and k3t in Figure 2. Please remove the sentences “It is evident that … general trend.” (line 183-185) and “A decrease … other regions.“ (line 186 –187) in Figure 3. Also, please elaborate the results in Figure 3A.
- The conclusion is well explained. However, my suggestion is to add the future recommendation to related authorities or agencies (for example Ministry of Health or HR) in the conclusion part.
- Finally, please ensure that the entire writing of this paper follows the guidelines or format given by the journal.
Reviewer 2 Report
Comments and Suggestions for Authors
See report as attached.

Reviewer 3 Report
Comments and Suggestions for Authors
The article aims to model the regional doctor needs in Kazakhstan using the functional principal component analysis (FPCA) method and to estimate them until 2033. Since the distribution of healthcare personnel is important both globally and locally and is not frequently seen in the literature, it also has originality in terms of the study method.
-The contributions of the study to the literature should be listed in the introduction section.
-Hyperparameter setting and validation details for LSTM should be added, such as epoch number, optimizer, loss function should be specified.
-The statistical significance of the FPCA + LSTM combination should be measured.
-A link is provided for the dataset used in the study, but it is quite difficult to access the dataset from here. It would be appropriate to provide the dataset used in the study as an excel file.
Reviewer 4 Report
Comments and Suggestions for Authors
The authors discussed an approach based on functional principal component analysis of data to identify patterns in the availability of health workers across different regions of Kazakhstan forecasting their needs over the next decade. To evaluate the forecasting performance of the model, they employed rolling origin cross-validation with an expanding window. The resulting scores were forecasted one year ahead using ARIMA and LSTM. LSTM showed higher accuracy compared to ARIMA. The use of the FPCA method allowed us to identify global and local trends in the dynamics of the number of doctors.
The authors wrote in the abstract that 'it is expected that up to 2033, the trend towards an increase in the number of doctors will continue in the Republic as a whole and in most regions..' I am wondering what is new? This is a pretty straightforward conclusion as more doctors are required as the population increases.
What assumptions are required to apply FPCA?
How the data set is collected? Please write the complete detail.
How many observations you have in the data set?
How are the basis functions selected?
Please improve the presentation of the figures.
All mathematical symbols must be written correctly. For example, what is meant by k1t mentioned on page 5, line 169?
Figure 3 lines are indistinguishable.
Why is PAE not reported in Table 1?
Which software is used for the analysis?
The future work should be discussed in the conclusion section.
More recent work on FDA should be added.
Round 2
Reviewer 4 Report
Comments and Suggestions for Authors
The authors discussed an approach based on functional principal component analysis of data to identify patterns in the availability of health workers across different regions of Kazakhstan forecasting their needs over the next decade. To evaluate the forecasting performance of the model, they employed rolling origin cross-validation with an expanding window. The resulting scores were forecasted one year ahead using ARIMA and LSTM. LSTM showed higher accuracy compared to ARIMA. The use of the FPCA method allowed us to identify global and local trends in the dynamics of the number of doctors.
The authors addressed all my previous comments.